# High-mobility, trap-free charge transport in conjugated polymer diodes

Mark Nikolka[1], Katharina Broch[1], John Armitage[1], David Hanifi[2], Peer J. Nowack [3], Deepak Venkateshvaran [1], Aditya Sadhanala[1], Jan Saska[4], Mark Mascal[4], Seok-Heon Jung[5], Jin-Kyun Lee[5], Iain McCulloch[6,7], Alberto Salleo[2] & Henning Sirringhaus [1]

Charge transport in conjugated polymer semiconductors has traditionally been thought to be limited to a low-mobility regime by pronounced energetic disorder. Much progress has recently been made in advancing carrier mobilities in field-effect transistors through developing low-disorder conjugated polymers. However, in diodes these polymers have to date not shown much improved mobilities, presumably reflecting the fact that in diodes lower carrier concentrations are available to fill up residual tail states in the density of states. Here, we show that the bulk charge transport in low-disorder polymers is limited by water-induced trap states and that their concentration can be dramatically reduced through incorporating small molecular additives into the polymer film. Upon incorporation of the additives we achieve space-charge limited current characteristics that resemble molecular single crystals such as rubrene with high, trap-free SCLC mobilities up to 0.2 cm$^2$/Vs and a width of the residual tail state distribution comparable to $k_BT$.

[1] Optoelectronics Group, Cavendish Laboratory, JJ Thomson Avenue, Cambridge CB3 0HE, UK. [2] Department of Material Science and Engineering, Stanford University, Stanford, CA 94305, USA. [3] Faculty of Natural Sciences, Imperial College London, Exhibition Road, London SW7 2AZ, UK. [4] Department of Chemistry, University of California Davis, Davis, CA 95616, USA. [5] Department of Polymer Science and Engineering, Inha University, Incheon 402-751, South Korea. [6] King Abdullah University of Science and Technology (KAUST), Kaust Solar Center (KSC), Thuwal 23955-6900, Saudi Arabia. [7] Department of Chemistry and Centre for Plastic, Imperial College London, Exhibition Road, London SW7 2AZ, UK. Correspondence and requests for materials should be addressed to M.N. (email: mn390@cam.ac.uk) or to H.S. (email: hs220@cam.ac.uk)

Organic semiconductors and conjugated polymers exhibit certain unique physical properties, such as large optical absorption coefficients and high photoluminescence quantum yields, which have made them attractive materials for a range of optoelectronic applications, such as organic light-emitting diode displays. Their charge transport properties, however, are not an inherent strength: the relatively weak van der Waals intermolecular interactions, the comparatively low chemical purity, and the presence of pronounced structural and energetic disorder, which are particularly prevalent in solution-processed conjugated polymer films, tend to limit their charge transport properties. Studies of charge transport are often performed on single-carrier, hole-only diodes. When using traditional conjugated polymers, such as polyphenylenevinylenes (PPVs), clean space charge-limited conduction (SCLC) can be observed. The current flowing through a polymer film of thickness $L$ increases approximately with the square of the applied voltage $V$, that is, follows Mott–Gurney's law:

$$J_{SCLC} = 9/8\epsilon_0\epsilon_r\mu_{SCLC}V^2/L^3, \qquad (1)$$

where $\epsilon_0$ and $\epsilon_r$ are the vacuum and relative dielectric permittivities. Deviations from the ideal square voltage dependence can be observed at high voltages and low temperatures and can be explained in terms of a concentration and electric field dependence of the SCLC mobility, $\mu_{SCLC}$, that reflects the energetic disorder broadening of the density of states (DOS) into a Gaussian DOS with a width $E_B$ that is typically much larger than $k_BT$, that is, on the order of 100–200 meV. In such disorder-dominated polymer systems, the hole mobility is concomitantly limited to relatively low values, typically $<10^{-3}$–$10^{-2}$ cm²/Vs. Trap states associated with chemical impurities may well be present, but their effect on the transport of holes is overshadowed by the effects of energetic disorder. Such chemical impurities are, however, present: water-related trap states, for example, limit strongly the transport of electrons, as opposed to holes[1].

In spite of the inherent challenges posed by energetic disorder, it has been possible over the past 20 years to improve the charge transport properties of organic semiconductors dramatically by exploring the wealth of chemical structures that are synthetically accessible. Particularly, in organic field-effect transistors mobilities have improved by several orders of magnitude[2–4], for example, in the so-called donor–acceptor copolymers with alternating electron-rich and electron-deficient units such as indacenodithiophene (IDT), diketopyrrolo-pyrrole (DPP), naphtalenediimide, cyclopentadithiophene, or isoindigo, where high field-effect mobilities >1 cm²/Vs[3,5–7] have been demonstrated. The nearly amorphous microstructure with only weak aggregation[8] and resulting large-area uniformity of many of these solution-processed polymers make them an ideal choice for large-area industrial processing[9,10]. These polymers owe their high performance in field-effect transistors (FETs) to a well-defined planar backbone conformation with little variation in torsion angle between the molecular units along the chain, resulting in a low degree of energetic disorder[11].

Achieving a high carrier mobility in an FET configuration may be considered relatively easy, because the carrier concentration at the interface between the accumulation layer and the gate dielectric is high, that is, on the order of $10^{18}$–$10^{19}$ cm⁻³, making it relatively easy to fill any residual trap or tail states. In SCLC diode measurements, on the other hand, the injected charge carrier concentration in the bulk is much smaller, that is, typically on the order of $10^{15}$–$10^{16}$ cm⁻³, and the charge transport properties are more likely to remain disorder or trap limited than in FET measurements. Indeed, SCLC diode measurements on several of these high-performance donor–acceptor polymers have

only yielded modest hole mobility values, rarely exceeding $10^{-3}$ cm²/Vs[12–14].

We have recently observed that water-induced trap states strongly limit the hole transport and the electrical stability of low-disorder donor–acceptor copolymer FETs and that the concentration of such water-related trap states can be reduced by the incorporation of certain small molecular additives into the polymer films[10]. The microstructure of many of these nearly amorphous or semi-crystalline polymers, such as indaceno-dithiophene-co-benzothiadiazole (IDT-BT), is believed to contain free volume or even nanometer-sized voids in the range of a few volume %[15,16], which, if occupied by water molecules, results in strong charge trapping[10,17]. We have demonstrated previously that certain molecular additives can fill and displace or passivate water molecules within the polymer's free volume. We have identified two specific mechanisms by which this trap passivation can occur. The additive can either be a solid molecule, such as tetracyanoquinodimethane (TCNQ), which gets incorporated into the free volume and due to its strongly electron-withdrawing nitrile groups is able to interact with water molecules. The resulting interaction, which can be a displacement, binding, or chemical reaction, hence prevents water from forming trap states[18]. Alternatively, the additive can be (residual) solvents that remove water molecules altogether by azeotrope formation[18]. We have found that molecular additives that fall under the above categories are able to significantly enhance the device performance, operational stability, uniformity, and contact resistance of a wide range of high-mobility conjugated polymer FETs[10,19]. In particular, solvents, such as 1,2-dichlorobenzene (DCB), which form azeotropes with water and by their very nature are highly miscible with the polymer, have resulted in significant improvements of FET device characteristics.

Here, we would like to address the question whether the use of such additives for reducing the water-related trap concentration in FETs could also lead to an improvement in the so far somewhat disappointing bulk charge transport properties of these polymers in diodes.

## Results

**High-mobility, trap-free polymer diodes**. We fabricated hole-only diodes for a range of high-mobility conjugated polymers and investigated their SCLC characteristics using a small area (20 µm × 200 µm) crossbar diode structures with gold contacts with and without a molecular additive (Fig. 1a). Motivated by our studies of molecular additives in FETs[10], we incorporate the molecular additive into the films by spinning the films from a high boiling solvent (DCB) and subjecting the films only to a very short, low-temperature anneal (90 °C for 10 s), which, according to our previous work, leaves a few % by volume of residual solvent additives in the film. The residual solvent hence acts as an additive. We compare these "additive devices" to "devices without additive" that are subjected to either longer annealing at 90 °C for 1 h or extended storage in vacuum, which both removes the residual solvent from the film (Fig. 1b, c). We first present results for poly[[2,5-bis(2-octadecyl)-2,3,5,6-tetrahydro-3,6-diketo-pyrrolo[3,4-c]pyrrole-1,4-diyl]-alt-(2-octylnonyl)-2,1,3-benzo-triazole] (DPP-BTz), a semi-crystalline donor–acceptor polymer with a high field-effect mobility of 2 cm²/Vs in spin-coated films and up to 6 cm²/Vs in uniaxially aligned films[3,20]. We selected it here because it exhibits a high degree of face-on packing with respect to the substrate, which should favor a high carrier mobility for bulk transport along the out-of-plane direction. The device without additive (blue curve in Fig. 1d) exhibits a low current density and the observed voltage dependence is significantly stronger than what is expected from Mott–Gurney's law

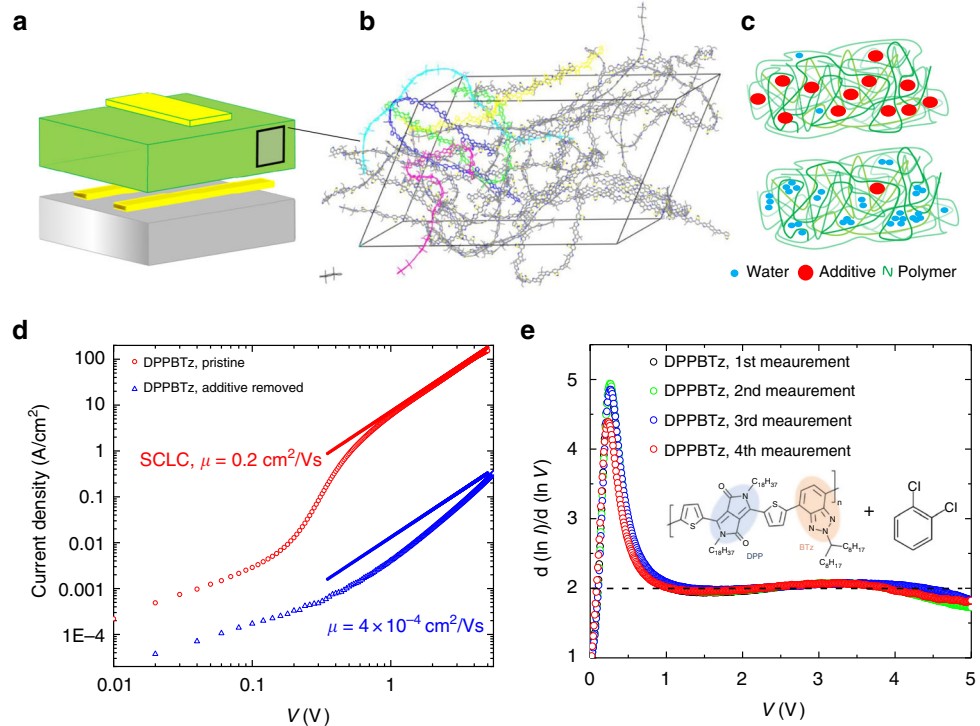

**Fig. 1** Space charge-limited conduction (SCLC) characteristics of poly[[2,5-bis(2-octadecyl)-2,3,5,6-tetrahydro-—3,6-diketopyrrolo[3,4-c]pyrrole-1,4-diyl]-alt-(2-octylnonyl)-2,1,3-benzotriazole] (DPP-BTz) diodes with and without additives. **a** Device structure of SCLC polymer diodes used (dimensions of crossbar overlap $A = 20 \times 200 \ \mu m^2$); **b** simulated structure of a high-mobility amorphous polymer (adapted from ref. [11]); **c** passivation of water-induced traps in the polymer's microstructure through the use of additives (red). Removal of additives allows water molecules (blue) to interact with polarons on the polymer backbone and disturb charge transport; **d** hole-only SCLC diode characteristics for the polymer DPP-BTz (thickness 220 nm). Using a solvent additive, record, field-independent SCLC mobilities of 0.2 cm²/Vs are reached. After solvent removal, performance drops by a factor of ~500 and trap-free SCLC behavior vanishes. **e** Power law exponent $m = \frac{\partial \mathrm{Ln}(I)}{\partial \mathrm{Ln}(V)}$ of the SCLC characteristics extracted during repeated measurements with excellent reproducibility. The chemical structure of DPP-BTz as well as the additive (residual 1,2-dichlorobenzene (DCB) solvent) is also shown

(indicated by the blue solid line). In the absence of a molecular additive (residual solvent DCB), the transport remains trap limited across the entire applied voltage range. In contrast, the additive device exhibits a much higher current density and a textbook-like SCLC behavior (Supplementary Note 1, Supplementary Fig. 1) with an Ohmic regime at low voltage, a steep trap-filling regime between 0.2 and 0.6 V, and a trap-free, SCLC regime above 1 V in which the J–V characteristics follow Mott–Gurney's law and a very high, field-independent SCLC mobility of 0.2 cm²/Vs is extracted (Fig. 1d, red line). Such a high-mobility value has never been reported for a diode based on a solution-processed conjugated polymer film, and even for small molecular single crystals such values remain unreached[21,22]. This mobility is within an order of magnitude of the FET mobility of DPP-BTz. To the best of our knowledge, such SCLC characteristics with distinct trap-filling and high-mobility trap-free regimes have not been observed in a conjugated polymer film before and resemble, in fact, the SCLC characteristics of molecular single crystals, such as rubrene[21,23] or pentacene[17].

Over the course of several hours, our devices exhibit excellent stability, and even during a full temperature cycle from 300 to 160 K, degradation was minimal (Supplementary Note 2, Supplementary Fig. 2; over longer periods (weeks) the beneficial effect of the solvent additive (DCB) is, however, gradually lost as the solvent evaporates from the films). This excellent stability is reflected when monitoring the power $m$ of the current density dependence on voltage (extracted through $m = \frac{\partial \ln(I)}{\partial \ln(V)}$) over repeated measurements (Fig. 1e, Supplementary Fig. 3a). Our diodes indeed exhibit robustly an ideal textbook-like SCLC

behavior, transitioning from a resistive regime ($m = 1$), through a steep trap-filling regime ($m \gg 2$) to an extended trap-free SCLC regime following the Mott–Gurney Law ($m = 2$).

**Extracting the width and distribution of traps.** We have investigated the temperature dependence of the J–V characteristics of DPP-BTz SCLC diodes with solvent additive (DCB solvent) in the temperature range from 300 to 160 K (Fig. 2a). The measured characteristics exhibited excellent reproducibility, with three consecutive temperature runs resulting in indistinguishable characteristics. From these data we can extract information on the width of the residual trap distribution: in the presence of an exponential distribution of trap states, $\rho_{trap} = \frac{N_t}{E_B} e^{-E/E_B}$ as a band tail beyond a mobility edge[22], with $N_t$ being the total trap density and $E_B$ the characteristic energy governing the width of the trap distribution, the SCLC characteristics can be expressed as:

$$J_{SCLC} = N_C q \mu \left( \frac{\varepsilon_0 \varepsilon_r}{q N_t e^{E_B/k_B T}} \right)^r \left[ \left( \frac{2r+1}{r+1} \right)^{r+1} \left( \frac{r}{r+1} \right)^r \right] \frac{V^{r+1}}{L^{2r+1}},$$

(2)

where $N_C$ is the effective DOS and $r$ is directly related to the width of the trap distribution according to $r = E_B/k_B T$. Typically, it is assumed that $E_B > k_B T$, which implies that $m = r + 1$ exceeds the ideal value $m = 2$ from the Mott–Gurney law. We applied Eq. 2 to the characteristics right above the trap-filling domain identified by an extended plateau with a constant slope. In this regime, the extracted width of the trap distribution has been shown to agree very well with a range of alternative models, such

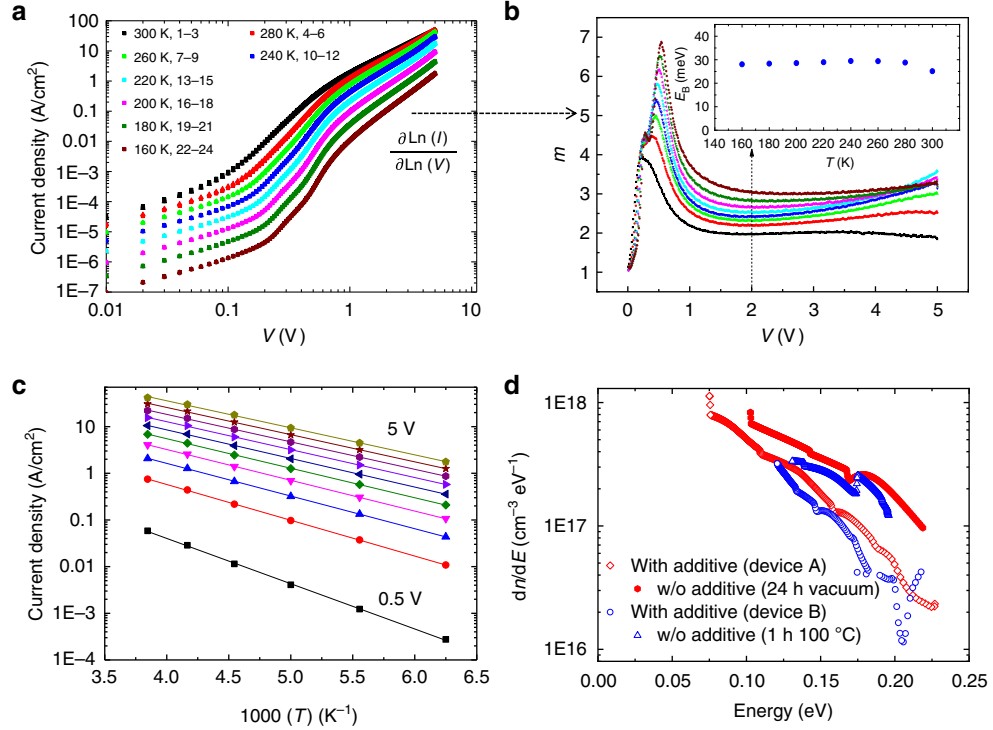

**Fig. 2** Density of trap states in poly[[2,5-bis(2-octadecyl)-2,3,5,6-tetrahydro-—3,6-diketopyrrolo[3,4-c]pyrrole-1,4-diyl]-alt-(2-octylnonyl)-2,1,3-benzotriazole] (DPP-BTz) space charge-limited conduction (SCLC) diodes. **a** Log–Log J–V characteristics of a DPP-BTz diode ($d = 220$ nm) with a solvent additive (1,2-dichlorobenzene (DCB)) measured at temperatures between 300 and 160 K; each characteristics was recorded three times to exclude degradation. **b** Corresponding power law exponent, $m = \frac{\partial \mathrm{Ln}(I)}{\partial \mathrm{Ln}(V)}$, of the SCLC characteristics as a function of temperature. The colors of the individual curves correspond to those in **a**. The width of the trap distribution ($E_B$) is estimated from Eq. 2 from the extended plateau (2 V) and is shown as an inset for all temperatures. **c** Arrhenius-type temperature activation of the current density for voltages between 0.5 and 5 V. **d** d$n$/d$E$ values extracted for DPP-BTz devices with and without an additive. The d$n$/d$E$ is shown for two separate devices (device A with a thickness of 170 nm and device B with a thickness of 220 nm), with the additive removed by annealing (blue) or by extended storage in vacuum (red)

as temperature-dependent SCLC (TD-SCLC) spectroscopy discussed later in this paper as well as more complex models taking into account drift and diffusion currents and a certain asymmetry of the charge injecting and extracting contacts[22]. From our data, we observe a clear increase of the slope in the extended plateau region between 1.5 and 3 V corresponding to an increase of the value of $r$ from 0.99 at 300 K to 2.08 at 160 K (Fig. 2b). We should note that at voltages higher than 3 V this trend still holds; however, in the intermediate temperature range, the results at high fields are affected by heating (Supplementary Note 2, Supplementary Figs. 3b, 4). Interestingly, when we multiply our values of $r$ with $T$, we obtain a temperature invariant value of $E_B = (28 \pm 3)$ meV (Fig. 2b, inset); this figure should be compared to values of 60–90 meV ($r = 2$–3) for materials such as poly[2-methoxy-5-(2-ethylhexyloxy)-1,4-phenylenevinylene] (MEH-PPV) (Supplementary Figs. 19, 20)[1]. This result suggests that for DPP-BTz diodes with the additive, the width of the residual trap distribution is on the same order as $k_B T$ and hence is narrow enough to exhibit very similar behavior to single crystalline semiconductors such as rubrene[24,25].

In single crystalline SCLC devices, the method of TD-SCLC spectroscopy[26,27] has been used to extract information on the trap state distribution (Supplementary Note 3, Supplementary Figs. 5–10). This method involves determining the activation energy of the current density from an Arrhenius fit at each applied voltage (Fig. 2c). It is therefore far more accurate than the previously applied method of extracting $E_B$ since instead of giving a snapshot of the trap distribution at a single voltage, it accounts for the entire voltage range. With increasing voltage, the activation energy is reduced from $E_A = 200$ meV (0.5 V) to $E_A$

$= 110$ meV (5 V) where it saturates (Supplementary Fig. 6); the latter value is in excellent agreement with the activation energy $E_A = 100$–110 meV of the FET mobility of DPP-BTz[3]. The method allows extracting the increment of the space charge with respect to the shift of the Fermi energy (d$n$/d$E$), which is directly related to the DOS through a convolution with the Fermi function. In this work, we chose to nevertheless only present d$n$/d$E$ values as these are directly obtained from the raw data and were found to differ little from the DOS values (Supplementary Note 3, Supplementary Figs. 7–8). DPP-BTz films with additive exhibit a significantly steeper d$n$/d$E$ dependence with order of magnitude fewer trap states over the entire band-tail region than devices without additive (Fig. 2d). We would like to highlight that this result is highly reproducible across several devices, and furthermore, it does not seem to affect d$n$/d$E$ (and hence, the DOS) if the solvent additive is removed from the polymer through annealing or slow evaporation in vacuum (Fig. 2d). The overall trap density obtained by integrating the area under the d$n$/d$E$ curves is, furthermore, in excellent agreement with trap densities extracted from an analysis of the trap-filling voltage $V_{TFL}$ according to $N_t = 3\varepsilon\varepsilon_0 V_{TFL}/2eL^2$. This analysis can only be done for devices with an additive exhibiting a clearly identifiable trap-filling domain. For DPP-BTz the analysis results in a trap density of $N_t = 0.8 \times 10^{-16}$/cm$^{-3}$, which should be compared to an area under the d$n$/d$E$ curves of $1 \times 10^{-16}$/cm$^{-3}$, which independently confirms the validity and accuracy of our DOS extraction. The extracted d$n$/d$E$ curves for devices with and without a solvent additive, furthermore, appear to converge close to $E = 0$, that is, the mobility edge. From these findings, we conclude that the presence of water in the polymer film induces a

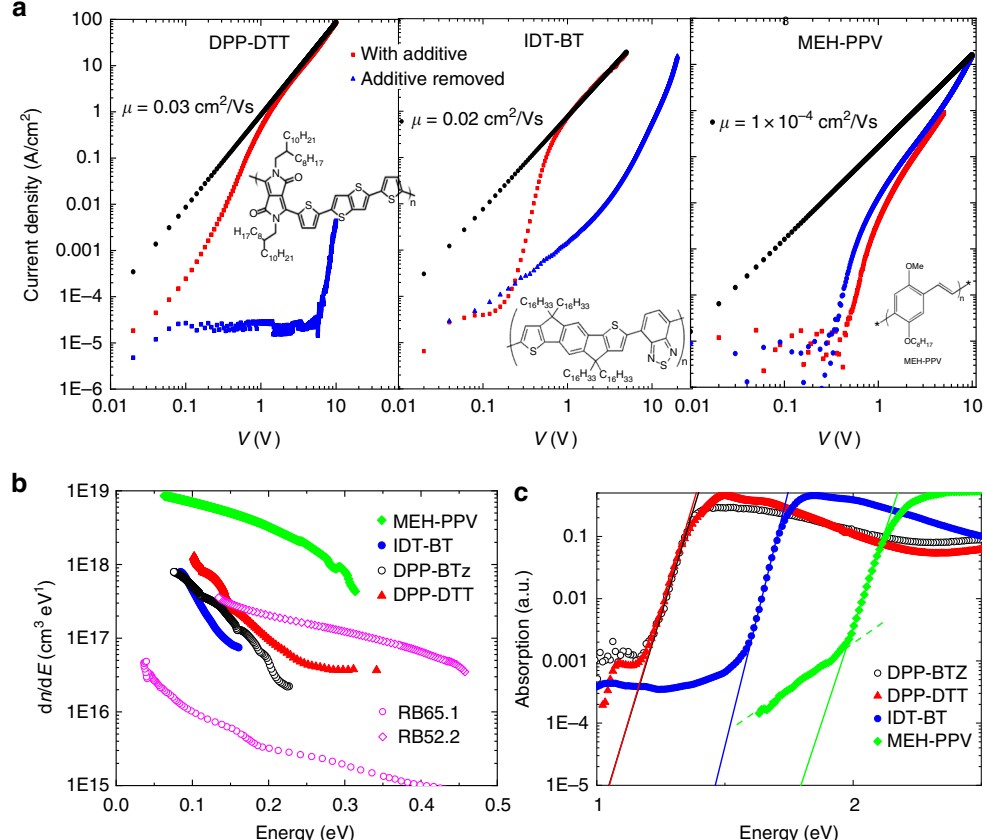

**Fig. 3** Trap removal through solvent additives in a range of polymers. **a** Log–Log J–V characteristics for the polymers diketopyrrolo-pyrrole-dithiophene-thienothiophene (DPP-DTT) (d = 130 nm, left panel), indacenodithiophen-co-benzothiadiazole (IDT-BT) (d=220 nm, central panel), and poly[2-methoxy-5-(2-ethylhexyloxy)-1,4-phenylenevinylene] (MEH-PPV) (d = 60 nm, right panel). Red squares correspond to devices with a solvent additive (residual 1,2-dichlorobenzene (DCB) solvent), blue triangles correspond to the same device after removal of the solvent additive, and black circles are fits to the space charge-limited conduction (SCLC) region with corresponding mobilities shown. The polymer's molecular structures are shown as an inset. **b** Density of trap states extracted from SCLC and optical spectroscopy: dn/dE values extracted using temperature-dependent SCLC spectroscopy for the polymers MEH-PPV, IDT-BT, poly[[2,5-bis(2-octadecyl)-2,3,5,6-tetrahydro-—3,6-diketopyrrolo[3,4-c]pyrrole-1,4-diyl]-alt-(2-octylnonyl)-2,1,3-benzotriazole] (DPP-BTz) and DPP-DTT, as well as for single crystals of rubrene (data from ref. [23]); all polymer data was obtained for devices with a solvent additive in place. **c** Absorption coefficient of IDT-BT, DPP-DTT, DPP-BTz, and MEH-PPV films, measured by photothermal deflection spectroscopy (PDS). Solid lines represent exponential tail fits for extraction of the Urbach energies

narrow, but continuous distribution of hole traps, as opposed to a discrete trap level, as was found for oxygen-related traps in rubrene[10,23] and that these can be effectively suppressed/passivated by the molecular additive.

Very similar results were obtained in a range of other high-mobility donor–acceptor copolymers with low disorder (Fig. 3a)[11], most notably the high-mobility polymer IDT-BT and diketopyrrolo-pyrrole-dithiophene-thienothiophene (DPP-DTT). We have also investigated MEH-PPV as a low-mobility reference system with higher energetic disorder. In IDT-BT and DPP-DTT, the removal of water traps through a solvent additive (residual DCB solvent also in this case) has a similarly dramatic effect on the J–V characteristics as discussed above for DPP-BTz (Supplementary Note 4, Supplementary Figs. 11–18). In devices without additives, the current density is up to 5 orders of magnitude lower than in devices with additive. The additive devices of IDT-BT and DPP-DTT exhibit similar SCLC behavior as DPP-BTz, with a trap-free Mott–Gurney regime at high fields, from which SCLC mobilities of 0.02 and 0.03 cm²/Vs, respectively, can be extracted. We should note, however, that IDT-BT exhibited a slight injection barrier evident by a lack of Ohmic behavior at low voltages (V < 0.1 V). A similar temperature-dependent analysis as conducted for DPP-BTz, furthermore, exhibits values for $E_B$ that are also on the

order of $k_B T$ for both DPP-DTT and IDT-BT (Supplementary Note 4).

Interestingly, in stark contrast, the SCLC characteristics of MEH-PPV cannot be improved significantly by using a solvent additive, and performance is very similar to device data published previously (Supplementary Note 4–5, Supplementary Figs. 19–23). It appears that in MEH-PPV the inherent energetic disorder of the polymer due to spatial variations in polymer chain conformations is so large that the transport properties remain strongly limited by this energetic disorder[1] and are less sensitive to trap states caused by chemical impurities such as water. This is not unexpected if the water-induced trap state distribution is relatively shallow, as suggested above. MEH-PPV has the highest tail state density as extracted by the dn/dE analysis (Fig. 3b, Supplementary Note 5, Supplementary Fig. 24) and exhibits only a weak decay into the band tail, and this is believed to reflect the pronounced inherent energetic disorder in this polymer. In contrast, the high-mobility polymers all exhibit a trap density that is significantly lower than for MEH-PPV, with a sharp exponential decay into the band tail (Supplementary Note 6–7, Supplementary Figs. 25–30). Interestingly, when we fit the exponential tail of the extracted DOS to extract a characteristic tail energy $E_{DOS}$ (Table 1), the values agree remarkably well with

**Table 1 Comparison of electronically and optically measured DOS**

| Polymer | $E_B$ (meV) | $E_u$ (meV) | $E_{DOS}$ (meV) | $\mu_{SCLC}$ (cm²/Vs) | $\mu_{FET}$ (cm²/Vs)[11,20] |
|---------|---------|---------|-----------|------------------|------------------------|
| MEH-PPV | >60 | 39 ± 2 | 60 ± 10 | 0.0001 | 0.0001 |
| IDT-BT | 24 | 24 ± 2 | 23 ± 3 | 0.02 | 2 |
| DPP-DTT | 30 | 31 ± 2 | 39 ± 5 | 0.03 | 1 |
| DPP-BTz | 28 | 32 ± 2 | 36 ± 5 | 0.2 | 2 |

Comparison of the width of the tail of trap states extracted from analysis of the *r* value ($E_B$), from an exponential fit of the d*n*/d*E* curves from SCLC spectroscopy ($E_{DOS}$) and the Urbach energies extracted from optical absorption spectroscopy ($E_u$) along with a comparison of FET and SCLC mobilities measured for the polymers MEH-PPV, IDT-BT, DPP-BTz, and DPP-DTT. The given errors were estimated from the uncertainty in the extraction procedure.
*FET* field-effect transistor, *MEH-PPV* poly[2-methoxy-5-(2-ethylhexyloxy)-1,4-phenylenevinylene], *IDT-BT* indacenodithiophene-*co*-benzothiadiazole, *DPP-BTz* poly[[2,5-bis(2-octadecyl)-2,3,5,6-tetrahydro-3,6-diketopyrrolo[3,4-*c*]pyrrole-1,4-diyl]-alt-(2-octylnonyl)-2,1,3-benzotriazole], *DPP-DTT* diketopyrrolo-pyrrole-dithiophene-thienothiophene, *SCLC* space charge-limited conduction, *DOS* density of states

the values of $E_B$ we extracted above from the measured *r* values in the trap-free SCLC regime. This confirms the consistency of our analysis. Only in the case of MEH-PPV we see a large uncertainty as values of $E_{DOS}$ depend strongly on which part of the DOS is fitted and likewise values of $E_B$ strongly depend on the voltage at which values are extracted.

**Comparison of electrically and optically measured trap distributions**. This difference in energetic disorder is also evident in the tail of the optical absorption of the different polymers. As an independent measure of energetic disorder, we have determined the Urbach energy[28] using photothermal deflection spectroscopy (PDS) (Fig. 3c). For IDT-BT, DPP-DTT, and DPP-BTz, we observe a sharp absorption onset translating into low Urbach energies of $E_U = 24$, 31, and 32 meV, respectively. These values are similar to the $E_B$ values extracted from the SCLC measurements (Table 1). This is somewhat surprising and may be accidental, because in one case the single-carrier DOS near the highest occupied molecular orbital is measured, while in the other case it is the joint excitonic DOS. In any case, MEH-PPV exhibits an Urbach energy of 39 meV, indicating a higher degree of energetic disorder, and in this case the extracted $E_B$ value is also significantly higher than the Urbach energy. We have also referenced our polymer data to d*n*/d*E* values published for two devices made from single crystalline rubrene[23] of different quality. It is striking that the low-quality rubrene crystals (RB52.2) show a higher trap density than our high-mobility polymers with a trap tail extending >0.4 eV beyond the mobility edge and trap levels that are higher throughout the entire band tail. Not surprisingly, the high-quality rubrene single crystal (RB65.1) shows the lowest trap density of all the materials investigated. It is fascinating, however, that the overall trap densities in our high-mobility polymers with molecular additives can be comparable to those of rubrene single crystals.

**Characterization of film microstructure**. Finally, we turn to a discussion of the trap-free SCLC mobility values of the different polymers. Despite similar energetic disorder and reported FET mobilities, the SCLC mobility value of DPP-BTz is an order of magnitude higher than those of DPP-DTT and IDT-BT, even in the presence of a solvent additive (DCB). To investigate the impact of morphology, we have performed grazing-incidence X-ray diffraction (GIWAXS) measurements and variable angle spectroscopic ellipsometry (VASE) measurements (Supplementary Note 8, Supplementary Figs. 31–33, Supplementary Table 1) on polymer films prepared under conditions identical to those used for diode fabrication. We fabricated films on gold-coated silicon wafers with identical polymer film thickness to the ones used for our SCLC diode studies. In accordance to results published elsewhere[29], IDT-BT exhibits a low level of crystallinity with backbones predominantly oriented face-on with respect to the substrate plane

(Fig. 4a), and this does not change noticeably when the DCB solvent additive is removed by annealing at 90 °C for 1 h (Fig. 4b). DPP-DTT exhibits a more crystalline microstructure with the polymer backbone predominantly stacking edge-on with respect to the substrate plane (Fig. 4c); in this case, we also cannot observe any significant changes in microstructure once the DCB additive is removed (Fig. 4d). Lastly, the microstructure of DPP-BTz differs significantly from DPP-DTT. The polymer exhibits an exceptionally well-defined face-on orientation of the polymer backbone (Fig. 4e, f). This suggests that the high out-of-plane SCLC mobility of DPP-BTz reflects the pronounced face-on backbone orientation facilitating out-of-plane transport. DPP-DTT, on the other hand, exhibits a lower SCLC mobility due to an edge-on molecular orientation impeding out-of-plane transport through insulating alkyl chains making it less efficient. Interestingly, despite the fundamentally different morphology of DPP-DTT and DPP-BTz, the differences in SCLC mobility are only a factor of 10, whereas a widening of the trap distribution as seen when solvent additives are removed has a much more drastic impact on the charge carrier mobility. Based on the preferential face-on orientation, we would have expected IDT-BT to perform similarly to DPP-BTz, rather than DPP-DTT; however, this is clearly not the case. We attribute this to the more amorphous microstructure of IDT-BT, which implies fewer aggregates and close-contact points for interchain charge transport in the vertical direction than in the more crystalline, face-on-oriented DPP-BTz.

**Electrical stability of polymer diodes with additives**. Despite providing a powerful tool for fundamental physical studies, solvent additives (such as residual DCB used in all cases in this work), nevertheless, tend to evaporate over extended time periods, resulting in a gradual loss of trap passivation, which is a limitation for technological applications. From a perspective of long-term stability solid additives would be preferable. Unfortunately, many of the solid small molecular additives at our disposal have shown a tendency to crystallize and cluster, leading to shorted devices. Furthermore, there are tight selection criteria for solid additives, requiring them to comprise electron-withdrawing groups (such as nitrile groups) that are able to interact with water molecules[10,18]. This needs to be combined with a sufficiently high solubility to allow the additives to be co-deposited with the polymer and to get incorporated into the free volume of the polymer film without undergoing crystallization and clustering. We have discovered that soluble small molecular additives, such as F4MCTCNQ (2,3,5,6-tetrafluoro-7-methoxycarbonyl,-7,8,8-tricyanoquinodimethane), which are solid at room temperature, meet these requirements and impart excellent device stability[30]. We fabricated SCLC diodes with DPP-BTz and a small amount (2 wt%) of F4MCTCNQ as solid additive and verified that these devices exhibit comparable trap-free SCLC characteristics to the devices with residual solvents (Fig. 1d) and that the inclusion of F4MCTCNQ does not lead to significant doping of DPP-BTz. We

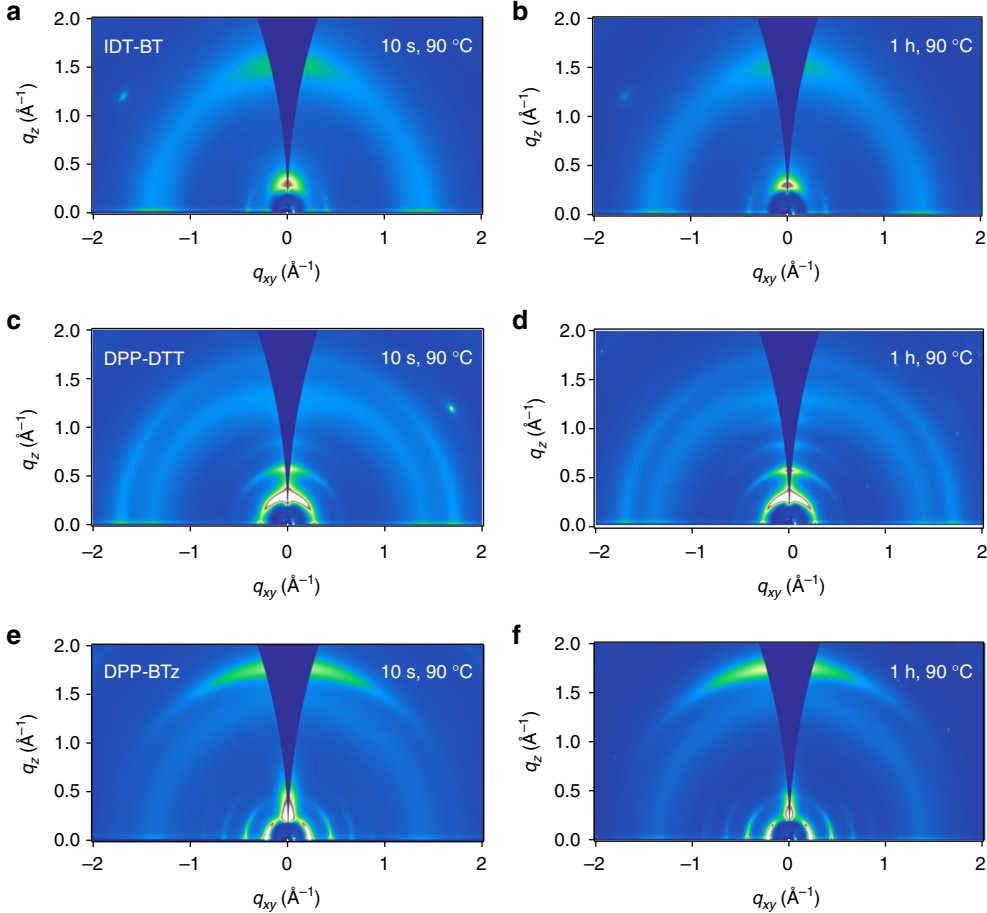

**Fig. 4** Microstructure of polymer films with and without a molecular additive grazing-incidence X-ray diffraction (GIWAXS) measurements of thick indacenodithiophene-*co*-benzothiadiazole (IDT-BT) (**a**, **b**), diketopyrrolo-pyrrole-dithiophene-thienothiophene (DPP-DTT) (**c**, **d**), and poly[[2,5-bis(2-octadecyl)-2,3,5,6-tetrahydro-3,6-diketopyrrolo[3,4-*c*]pyrrole-1,4-diyl]-alt-(2-octylnonyl)-2,1,3-benzotriazole] (DPP-BTz) (**e**, **f**) films on gold substrates; films with the additive (residual 1,2-dichlorobenzene (DCB) solvent) are shown in **a**, **c**, **e** and films with the additive removed are shown in **b**, **d**, **f**

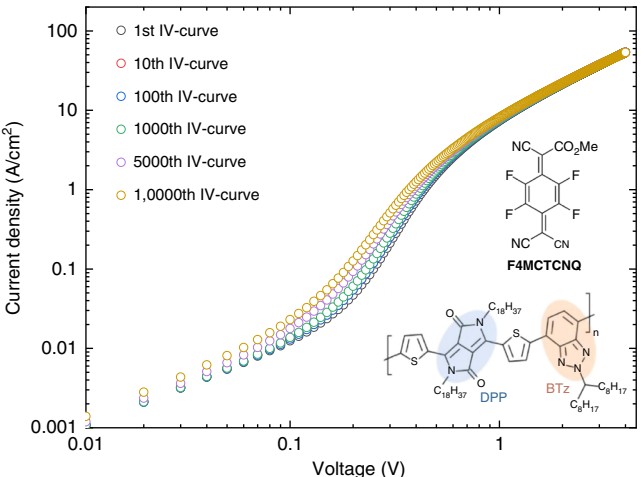

**Fig. 5** Stability of an space charge-limited conduction (SCLC) diode with a solid molecular additive stability of a poly[[2,5-bis(2-octadecyl)-2,3,5,6-tetrahydro-3,6-diketopyrrolo[3,4-*c*]pyrrole-1,4-diyl]-alt-(2-octylnonyl)-2,1,3-benzotriazole] (DPP-BTz) SCLC diode with 2 wt% of the highly soluble solid additive F4MCTCNQ (2,3,5,6-tetrafluoro-7-methoxycarbonyl,-7,8,8-tricyanoquinodimethane) during 10,000 backward and forward *I–V* cycles

tested the stability of the electrical characteristics under operational stress and found that devices with solid additives exhibit excellent stability with no degradation of the SCLC current occurring after 10,000 forward and backward measurement cycles (Fig. 5). Only a small increase in current and a voltage shift in the characteristics (<0.1 V) is observed in the trap-filling region after these 10,000 measurement cycles. The achieved stability is particularly impressive given the high current densities of almost 60 A/cm² that are reached in our diodes and, to the best of our knowledge, represents some of the most stable characteristics of organic diodes ever reported in the literature. This demonstrates that our water trap passivation technique not only provides significant enhancements in diode performance but may also offer a practical route to improving long-term stability of polymer diodes.

## Discussion

In summary, we have shown that record SCLC mobilities of up to 0.2 cm²/Vs can be obtained for bulk transport in solution-processed, low-disorder polymer diodes. In these polymers the width of the tail of the bulk DOS can be on the order of $k_BT$, and hence SCLC diodes exhibit characteristics akin to molecular single crystals with a distinct trap-filling regime and a (nearly) trap-free Mott–Gurney regime. Our results extend previous FET studies[11] and show that, not only at the interface, but also in the bulk of these solution-processed donor–acceptor polymer films, energetic disorder broadening of the DOS is approaching surprisingly low levels once water-related traps have been removed. Given that

their lower carrier concentrations render diodes far more susceptible to traps than FETs, our present work is indeed surprising. We therefore conclude that water-related traps are one of the main charge transport-limiting factors in high-mobility conjugated polymers at low carrier concentrations. Replacing solvent additives with highly soluble small molecular additives, we were furthermore able to extend the stability of our diodes and demonstrated devices with excellent long-term stability during 10,000 measurement cycles. Our work, therefore, constitutes an important proof-of-concept demonstration that trap-free transport properties, high carrier mobilities for bulk transport, and a narrow density of tail states near the band edges can be achieved in low-disorder conjugated polymers with the help of molecular additives. Our findings are likely to have important applications for improving not only the charge transport properties of polymer diodes, light-emitting diodes, and solar cells, but might also allow the improvement of other characteristics of these devices, including reducing trap-assisted non-radiative recombination or enhancing quasi-Fermi level splitting and open-circuit voltages.

## Methods

**Device fabrication**. SCLC diodes were fabricated on glass substrates with photolithographically defined bottom contacts of Ti/Au (10 nm/30 nm). Polymers were then deposited by spin coating from 20 g/l DCB solutions giving film thicknesses in the range of 130–220 nm. Residual DCB solvent was intentionally left in the film to act as an additive, and films were only annealed at 90 °C for 10 s to drive out bulk solvent. A 200-μm-wide and 20-nm-thick Au crossbar was subsequently evaporated at base pressures of $6 \times 10^{-6}$ mbar to form the diode's top contact. To avoid heating during the evaporation, which could lead to evaporation of the residual solvent from the films, an evaporation chamber with a large source-sample distance and water-cooled sample stage was used. The evaporation chamber was furthermore left to cool for 12 h before evaporating. To guarantee reproducibility, all fabrication steps and electrical measurements were performed in a $N_2$ glove box. The electrical measurements reported in this work were performed in vacuum in a Desert Cryogenics probe station. To make sure that there was good thermal contact between the measurement chuck and the device, we used a silicone- and halogen-free cryogenic high vacuum grease.

**Film thickness: atomic force microscopy**. To measure surface morphologies, a Vecco Dimension 3100 atomic force microscope (AFM) was used, with all images being recorded in tapping mode. The film thickness was recorded by making a thin scratch on top of the gold electrodes using a pair of tweezers while making sure that the underlying gold layer was not damaged. Subsequently, the AFM was scanned across the cut and the film thickness could thus be extracted from the resulting image.

**Photothermal deflection spectroscopy**. A home-built PDS setup was used to measure sub-band gap absorption spectra. The technique is based on the heat energy that is released from the surface of a sample when monochromatic light is absorbed. An inert liquid surrounding the sample dissipates this thermal energy, changes its refractive index, and as a result deflects a laser beam, which is sent at grazing incidence along the sample's surface. Using a quadrant detector connected to a lock-in amplifier, the deflection of the laser beam is recorded as a function of the monochromatic pump wavelength.

**VASE measurements**. The dielectric function was determined using VASE with a Woolam M2000 Spectroscopic Ellipsometer in the wavelength range from 900 to 400 nm. The angles were varied from 40° to 70° in steps of 5°. The film thickness was determined in the transparent range from 2000 to 1200 nm using a Cauchy function to describe the wavelength-dependent real part of the complex refractive index $\tilde{n}$, assuming that the extinction coefficient (imaginary part of $\tilde{n}$) is zero.

**GIWAX measurements**. GIWAXS measurements were performed on polymer films with and without a molecular additive deposited on top of a gold-coated silicon wafer. The same thicknesses and processing conditions as reported for the SCLC diodes were used. All GIWAX measurements were performed on the 11-3 beamline (two-dimensional (2D) grazing incidence wide angle with MAR225 image-plate area detector) at the Stanford Synchrotron Radiation Lightsource. The incident photon energy that was used was 12.732 keV (0.973 Å). For all measurements, He(g) environments were used to minimize air scatter and beam damage to samples. The 2D grazing-incidence sample-detector distance for the experiment was 315 mm calibrated with a polycrystalline lanthanide hexaboride standard at a 3.0° angle with respect to the critical angle of the calibrant. For grazing-incidence geometries, the incidence angle was set below the critical angle of 0.1° for the polymer film, which is above the critical angle of the underlying native oxide substrate (Table 1).

## Data availability

The data that support the findings of this study are available at the University of Cambridge data repository at https://doi.org/10.17863/CAM.36394.

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

## Acknowledgements

We would like to thank Prof. Cornelius Krellner for providing us with d$n$/d$E$ data on rubrene single crystals. We gratefully acknowledge financial support of the Engineering and Physical Sciences Research Council (EPSRC) through a Program Grant (EP/M005141/1). M.N. acknowledges financial support from the European Commission through a Marie-Curie Individual Fellowship (EC Grant Agreement Number: 747461).

## Author contributions

M.N. designed the study and performed and analyzed all electrical measurements; K.B. measured and analyzed VASE data; J.A. and D.V. contributed to sample fabrication and electrical stress measurements; A.S. measured PDS absorption in polymer thin films; D. H. and A.S. performed GIWAX measurements and contributed to discussions on data analysis; P.J.N. contributed to the data analysis by developing a ridge regression model for DOS extraction; J.S. and M.M. synthesized F4MCTCNQ; S.-H.J. and J.K.L. synthesized DPP-BTz; I.M. supplied IDT-BT; H.S. directed and coordinated the research; M.N., A.S., and H.S. wrote the manuscript.

## Additional information

**Competing interests:** The authors declare no competing interests.

