## [Peer Review File · Nature Communications]

Reviewers' comments:

Reviewer #1 (Remarks to the Author):

The authors report a significant finding that the space charge limited current in polymers can be substantially enhanced by the addition of molecular additives. The hypothesis is that these molecules fill traps sites caused by water, etc.

The data is well-presented and thorough. I have the following comments on the interpretation.

1) The authors refer to "voids" in the polymer film. Voids would usually refer to larger scale open spaces in bulk samples. I think the authors actually mean "free volume" which is common in all non-crystalline polymers. See reference below for more common descriptions of what free volume refers to in amorphous polymers:

Polymer Free Volume and Its Connection to the Glass Transition
Macromolecules, 2016, 49 (11), pp 3987–4007
DOI: 10.1021/acs.macromol.6b00215

2) The authors seem improvements with the addition of a solvent, DCB. Addition of solvents will depress the glass transition, T_g , of the polymer particularly at the relatively high percentages mentioned. Do the authors have DSC data on this? There is some possibility that this would also change the charge hopping dynamics as well.

3) At the higher current densities does the temperature of the device rise (devices are on glass)? The power looks to be $>100\text{W}/\text{cm}^2$ in some of the curves and reaches $1000\text{W}/\text{cm}^2$ in Figure 3a. This point also goes along with the question about the depression of T_g .

4) It is interesting that the addition of F4MCTCNQ does not lead to doping in the polymers under study. Do the authors know if charge transfer states are formed? There have been some recent papers showing CT states in the absence of integer charge transfer. This feature may be hard to notice without PDS spectra depending on the amount of the small molecule added. See:

Polymorphism controls the degree of charge transfer in a molecularly doped semiconducting polymer
DOI: 10.1039/C8MH00223A Mater. Horiz., 2018, 5, 655-660

Branched Side Chains Govern Counterion Position and Doping Mechanism in Conjugated Polythiophenes
ACS Macro Lett., 2018, 7, pp 1492–1497 DOI: 10.1021/acsmacrolett.8b00778

5) one might expect the F4MCTCNQ to change the injection barrier at the Au-electrode as well due to interaction of the small molecule with the metal. This should also be mentioned because it is possible that the two additives can function by different mechanisms.

Reviewer #2 (Remarks to the Author):

This is an outstanding paper describing a breakthrough in diode performance for polymer semiconductors. The authors have built in their previous findings about the role of water in high performance polymer FETs. They show here that molecular additives can passivate traps (presumably due to water) such that ideal SCLC behavior can be observed as was done in single crystal materials in the 1980s (Norbert Karl) and 1990s and early 2000s. The observation of a

sharp transition from ohmic conduction to SCLC is the hallmark of trap filling and it allows (1) cleaner extraction of the charge mobility and (2) access to the electronic density of states. The authors nicely compare the trap DOS they extract from the diode measurements with an independent measurement of traps by photothermal deflection spectroscopy. The work is self-consistent and is an important breakthrough. I have no significant criticisms.

Reviewer #3 (Remarks to the Author):

This paper reports on studies of the effects of additives on the electrical properties of diodes fabricated with polymeric semiconductor films. The results show that the concentration of water-induced trap states in the bulk of the semiconducting film can be reduced through the incorporation of small molecular additives, having a significant effect on the current-voltage characteristics of the diodes.

This work is an extension of previous studies that were carried-out on field-effect transistors. The claimed novelty of this paper is that the effects of such additives are studied at lower carrier concentrations through space-charge limited current measurements.

The paper is of high quality and provides a comprehensive study that yields important information of the properties of traps in polymeric semiconductors such as the width of the residual trap distribution. The authors clearly indicate how the conclusions in this study differs from those of previous studies. The claims of this new study advance the understanding of the charge transport properties of organic semiconductors and how material properties can be derived from SCLC measurements when the density of traps is reduced in polymers that have limited energetic disorder.

The SCLC measurements in these diodes when additives are added to the semiconductor yield curves that show three distinct regimes one would expect from such experiments. However, I was surprised that the authors did not discuss in the analysis of their data, the value of the threshold voltage for which the current changes from the linear regime to an SCLC regime. Such a discussion could provide a self-consistency check and would provide additional value to the paper.

We would like to thank all the reviewers for their constructive feedback. We are very delighted that all three reviewers consider our findings significant, novel and even outstanding. In our reply below, we have addressed the remaining technical points raised. We have also modified the manuscript accordingly.

Reviewer #1 (Remarks to the Author):

The authors report a significant finding that the space charge limited current in polymers can be substantially enhanced by the addition of molecular additives. The hypothesis is that these molecules fill traps sites caused by water, etc.

The data is well-presented and thorough. I have the following comments on the interpretation.

We thank the reviewer for their positive assessment and the constructive comments.

1) The authors refer to “voids” in the polymer film. Voids would usually refer to larger scale open spaces in bulk samples. I think the authors actually mean “free volume” which is common in all non-crystalline polymers. See reference below for more common descriptions of what free volume refers to in amorphous polymers:

Polymer Free Volume and Its Connection to the Glass Transition
Macromolecules, 2016, 49 (11), pp 3987–4007, DOI: 10.1021/acs.macromol.6b00215

It is true that what we call a “void” is the free volume in the polymer rather than a large-scale cavity. However, to a certain degree, this is a question of semantics. We used the term “void” here to highlight that the free volume is large enough to accommodate water as well as the molecular additives. Yet, the referee is right that this must be clarified. We now initially clarify that polymers contain free volume or even voids that are nanometre sized and use the term free volume throughout the rest of manuscript. We also added a reference to the paper the referee suggested.

2) The authors seem improvements with the addition of a solvent, DCB. Addition of solvents will depress the glass transition, T_g , of the polymer particularly at the relatively high percentages mentioned. Do the authors have DSC data on this? There is some possibility that this would also change the charge hopping dynamics as well.

For many of these polymers, it is impossible to detect a glass transition using DSC (the crystallinity is generally not high enough to pick this up). This is the also the reason why we performed GIWAXS measurements to exclude changes in morphology due to the presence of solvents. From these results (Figure 4) we are confident, that solvents do not lead to any significant change in the polymer's microstructure.

3) At the higher current densities does the temperature of the device rise (devices are on glass)? The power looks to be $>100\text{W}/\text{cm}^2$ in some of the curves and reaches $1000\text{W}/\text{cm}^2$ in Figure 3a. This point also goes along with the question about the depression of T_g .

The referee raises an important point. We have measured characteristics repetitively (3 times for every temperature) to make sure that traces overlap entirely and that no irreversible, heating related degradation occurs. To further exclude heating artefacts, we also fabricated devices on silicon which exhibited identical characteristics. The corresponding discussion on this topic can be found in the Supporting Information, Section 3. From these checks we are confident, that within the voltage ranges

we have chosen, the power dissipation does not lead to significant changes in either the overall microstructure or electrical performance.

Figure: (left) Comparison of DPP-BTz SCLC diode characteristics fabricated on silicon (black) and on glass (red); (right) corresponding derivative m of the I - V characteristics.

4) It is interesting that the addition of F4MCTCNQ does not lead to doping in the polymers under study. Do the authors know if charge transfer states are formed? There have been some recent papers showing CT states in the absence of integer charge transfer. This feature may be hard to notice without PDS spectra depending on the amount of the small molecule added. See:

Polymorphism controls the degree of charge transfer in a molecularly doped semiconducting polymer
DOI: 10.1039/C8MH00223A Mater. Horiz., 2018, 5, 655-660

Branched Side Chains Govern Counterion Position and Doping Mechanism in Conjugated Polythiophenes: ACS Macro Lett., 2018, 7, pp 1492–1497 DOI: 10.1021/acsmacrolett.8b00778

We cannot exclude entirely any charge transfer states between DPP-BTz and F4MCTCNQ, although the current at low Voltages do not change significantly under the presence of F4MCTCNQ. Generally, SCLC diodes will be very sensitive even to small changes in bulk conductivity, so we would still expect to see the occurrence of charge transfer. Furthermore, in our earlier work (*Nature Materials*, **16**, 356–362, 2017) we were able to show that in field-effect transistors, the electron affinity of the additive in fact, does not play a major role in improving device performance. Instead, we could demonstrate that this is related to the passivation of water related traps.

5) one might expect the F4MCTCNQ to change the injection barrier at the Au-electrode as well due to interaction of the small molecule with the metal. This should also be mentioned because it is possible that the two additives can function by different mechanisms.

Following from point 4), we have performed an extensive analysis on field-effect transistors (*Nature Materials*, **16**, 356–362, 2017) where we have demonstrated that additives such as Aminobenzonitrile (ABN) with electron affinities 2eV below the polymer's HOMO level, were able to improve injection and contact resistance (See attached Figure). We were also able to show that the small molecular additives are affecting bulk properties rather than only charge injection and could link this to the passivation of water traps. The prevailing mechanism is however, quite complex and in this work, we would like to focus on the impact of traps on SCLC diodes and the density of states only.

Figure: Transfer-Line Measurements to extract the contact resistance of IDT-BT field-effect transistors with various molecular additives.

Reviewer #2 (Remarks to the Author):

This is an outstanding paper describing a breakthrough in diode performance for polymer semiconductors. The authors have built in their previous findings about the role of water in high performance polymer FETs. They show here that molecular additives can passivate traps (presumably due to water) such that ideal SCLC behavior can be observed as was done in single crystal materials in the 1980s (Norbert Karl) and 1990s and early 2000s. The observation of a sharp transition from ohmic conduction to SCLC is the hallmark of trap filling and it allows (1) cleaner extraction of the charge mobility and (2) access to the electronic density of states. The authors nicely compare the trap DOS they extract from the diode measurements with an independent measurement of traps by photothermal deflection spectroscopy. The work is self-consistent and is an important breakthrough. I have no significant criticisms.

We would like to thank the reviewer for such a strong endorsement of our work. We are extremely delighted about this positive review and feedback.

Reviewer #3 (Remarks to the Author):

This paper reports on studies of the effects of additives on the electrical properties of diodes fabricated with polymeric semiconductor films. The results show that the concentration of water-induced trap states in the bulk of the semiconducting film can be reduced through the incorporation of small molecular additives, having a significant effect on the current-voltage characteristics of the diodes.

This work is an extension of previous studies that were carried-out on field-effect transistors. The claimed novelty of this paper is that the effects of such additives are studied at lower carrier concentrations through space-charge limited current measurements.

The paper is of high quality and provides a comprehensive study that yields important information of the properties of traps in polymeric semiconductors such as the width of the residual trap distribution. The authors clearly indicate how the conclusions in this study differs from those of previous studies.

The claims of this new study advance the understanding of the charge transport properties of organic semiconductors and how material properties can be derived from SCLC measurements when the density of traps is reduced in polymers that have limited energetic disorder.

We thank the referee for this assessment and are very glad that our paper was perceived as high quality.

The SCLC measurements in these diodes when additives are added to the semiconductor yield curves that show three distinct regimes one would expect from such experiments. However, I was surprised that the authors did not discuss in the analysis of their data, the value of the threshold voltage for which the current changes from the linear regime to an SCLC regime. Such a discussion could provide a self-consistency check and would provide additional value to the paper.

This is a very justified comment. We have indeed not included this analysis in our paper and at the time of writing only did a back of the envelope estimate as a check. The referee has got an excellent point though, that this would provide a good self-consistency check and indeed add value to the analysis. For the device shown in Figure 1 we have exemplarily calculated the trap density from the threshold voltage V_{TFL} (1.3 V) according to:

$$N_t = \frac{3\epsilon\epsilon_0 V_{TFL}}{2qd^2} = 8 \cdot 10^{-15} \text{ cm}^{-3}$$

This should be compared to the area under the dn/dE curve (Fig. 2) which results in a value of $1 \cdot 10^{-16}$. The slight discrepancy between these values can be expected to be within the error margin of both analysis (especially e.g. accurate determination of V_{TFL}) and clearly shows that the methods are consistent. We have now added a short discussion on this analysis to the manuscript.

REVIEWERS' COMMENTS:

Reviewer #1 (Remarks to the Author):

The authors have clearly addressed the comments of the reviews. Excellent manuscript and nice to see a SCLC paper about something other than MEH-PPV!

Reviewer #2 (Remarks to the Author):

I am satisfied with the authors' responses to the collective reviewers' comments.

I will note, though, that there seems to be some confusion about the meaning of the glass transition temperature in the authors' response to reviewer 1. The glass transition applies to amorphous domains not to the crystalline domains (T_{melt} is always higher than T_g as well).

In general, I agree that it is difficult to detect T_g for many electronic polymers...there is more here that the community needs to address.

Whether additives impacts T_g is an open question, and how this impacts transport is also open, but I am still comfortable with the important contribution of this current paper.

Reviewer #3 (Remarks to the Author):

Revised version addressed my comments and concerns.